# Quantized Estimation of Gaussian Sequence Models in Euclidean Balls

**Yuancheng Zhu**      **John Lafferty**

Department of Statistics
University of Chicago

## Abstract

A central result in statistical theory is Pinsker's theorem, which characterizes the minimax rate in the normal means model of nonparametric estimation. In this paper, we present an extension to Pinsker's theorem where estimation is carried out under storage or communication constraints. In particular, we place limits on the number of bits used to encode an estimator, and analyze the excess risk in terms of this constraint, the signal size, and the noise level. We give sharp upper and lower bounds for the case of a Euclidean ball, which establishes the Pareto-optimal minimax tradeoff between storage and risk in this setting.

## 1   Introduction

Classical statistical theory studies the rate at which the error in an estimation problem decreases as the sample size increases. Methodology for a particular problem is developed to make estimation efficient, and lower bounds establish how quickly the error can decrease in principle. Asymptotically matching upper and lower bounds together yield the minimax rate of convergence

$$R_n(\mathcal{F}) = \inf_{\widehat{f}} \sup_{f \in \mathcal{F}} R(\widehat{f}, f).$$

This is the worst-case error in estimating an element of a model class $\mathcal{F}$, where $R(\widehat{f}, f)$ is the risk or expected loss, and $\widehat{f}$ is an estimator constructed on a data sample of size $n$. The corresponding sample complexity of the estimation problem is $n(\epsilon, \mathcal{F}) = \min\{n : R_n(\mathcal{F}) < \epsilon\}$.

In the classical setting, the infimum is over all estimators. In contemporary settings, it is increasingly of interest to understand how error depends on computation. For instance, when the data are high dimensional and the sample size is large, constructing the estimator using standard methods may be computationally prohibitive. The use of heuristics and approximation algorithms may make computation more efficient, but it is important to understand the loss in statistical efficiency that this incurs. In the minimax framework, this can be formulated by placing computational constraints on the estimator:

$$R_n(\mathcal{F}, B_n) = \inf_{\widehat{f} : C(\widehat{f}) \leq B_n} \sup_{f \in \mathcal{F}} R(\widehat{f}, f).$$

Here $C(\widehat{f}) \leq B_n$ indicates that the computation $C(\widehat{f})$ used to construct $\widehat{f}$ is required to fall within a "computational budget" $B_n$. Minimax lower bounds on the risk as a function of the computational budget thus determine a feasible region for computation-constrained estimation, and a Pareto-optimal tradeoff for error versus computation.

One important measure of computation is the number of floating point operations, or the running time of an algorithm. Chandrasekaran and Jordan [3] have studied upper bounds for statistical estimation with computational constraints of this form in the normal means model. However, useful lower bounds are elusive. This is due to the difficult nature of establishing tight lower bounds for

this model of computation in the polynomial hierarchy, apart from any statistical concerns. Another important measure of computation is storage, or the space used by a procedure. In particular, we may wish to limit the number of bits used to represent our estimator $\widehat{f}$. The question then becomes, how does the excess risk depend on the budget $B_n$ imposed on the number of bits $C(\widehat{f})$ used to encode the estimator?

This problem is naturally motivated by certain applications. For instance, the Kepler telescope collects flux data for approximately 150,000 stars [6]. The central statistical task is to estimate the lightcurve of each star nonparametrically, in order to denoise and detect planet transits. If this estimation is done on board the telescope, the estimated function values may need to be sent back to earth for further analysis. To limit communication costs, the estimates can be quantized. The fundamental question is, what is lost in terms of statistical risk in quantizing the estimates? Or, in a cloud computing environment (such as Amazon EC2), a large number of nonparametric estimates might be constructed over a cluster of compute nodes and then stored (for example in Amazon S3) for later analysis. To limit the storage costs, which could dominate the compute costs in many scenarios, it is of interest to quantize the estimates. How much is lost in terms of risk, in principle, by using different levels of quantization?

With such applications as motivation, we address in this paper the problem of risk-storage tradeoffs in the normal means model of nonparametric estimation. The normal means model is a centerpiece of nonparametric estimation. It arises naturally when representing an estimator in terms of an orthogonal basis [8, 11]. Our main result is a sharp characterization of the Pareto-optimal tradeoff curve for quantized estimation of a normal means vector, in the minimax sense. We consider the case of a Euclidean ball of unknown radius in $\mathbb{R}^n$. This case exhibits many of the key technical challenges that arise in nonparametric estimation over richer spaces, including the Stein phenomenon and the problem of adaptivity.

As will be apparent to the reader, the problem we consider is intimately related to classical rate distortion theory [7]. Indeed, our results require a marriage of minimax theory and rate distortion ideas. We thus build on the fundamental connection between function estimation and lossy source coding that was elucidated in Donoho's 1998 Wald Lectures [4]. This connection can also be used to advantage for practical estimation schemes. As we discuss further below, recent advances on computationally efficient, near-optimal lossy compression using sparse regression algorithms [12] can perhaps be leveraged for quantized nonparametric estimation.

In the following section, we present relevant background and give a detailed statement of our results. In Section 3 we sketch a proof of our main result on the excess risk for the Euclidean ball case. Section 4 presents simulations to illustrate our theoretical analyses. Section 5 discusses related work, and outlines future directions that our results suggest.

## 2   Background and problem formulation

In this section we briefly review the essential elements of rate-distortion theory and minimax theory, to establish notation. We then state our main result, which bridges these classical theories.

In the rate-distortion setting we have a source that produces a sequence $X^n = (X_1, X_2, \ldots, X_n)$, each component of which is independent and identically distributed as $\mathcal{N}(0, \sigma^2)$. The goal is to transmit a realization from this sequence of random variables using a fixed number of bits, in such a way that results in the minimal expected distortion with respect to the original data $X^n$. Suppose that we are allowed to use a total budget of $nB$ bits, so that the average number of bits per variable is $B$, which is referred to as the *rate*. To transmit or store the data, the *encoder* describes the source sequence $X^n$ by an index $\phi_n(X^n)$, where

$$\phi_n : \mathbb{R}^n \to \{1, 2, \ldots, 2^{nB}\} \equiv \mathbb{C}(B)$$

is the *encoding function*. The $nB$-bit index is then transmitted or stored without loss. A *decoder*, when receiving or retrieving the data, represents $X^n$ by an estimate $\check{X}^n$ based on the index using a *decoding function*

$$\psi_n : \{1, 2, \ldots, 2^{nB}\} \to \mathbb{R}^n.$$

The image of the decoding function $\psi_n$ is called the *codebook*, which is a discrete set in $\mathbb{R}^n$ with cardinality no larger than $2^{nB}$. The process is illustrated in Figure 1, and variously referred to as

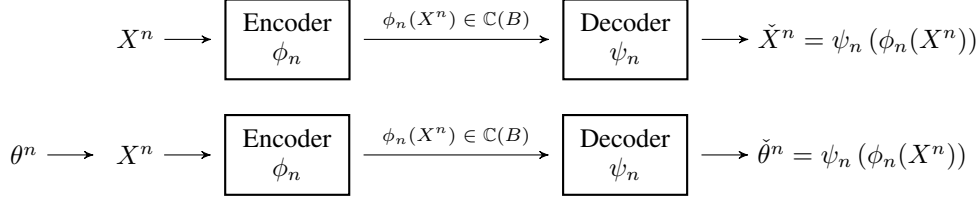

Figure 1: Encoding and decoding process for lossy compression (top) and quantized estimation (bottom). For quantized estimation, the model (mean vector) $\theta^n$ is deterministic, not random.

source coding, lossy compression, or quantization. We call the pair of encoding and decoding functions $Q_n = (\phi_n, \psi_n)$ an $(n, B)$-*rate distortion code*. We will also use $Q_n$ to denote the composition of the two functions, i.e., $Q_n(\cdot) = \psi_n(\phi_n(\cdot))$.

A *distortion measure*, or a *loss function*, $d : \mathbb{R} \times \mathbb{R} \to \mathbb{R}^+$ is used to evaluate the performance of the above coding and transmission process. In this paper, we will use the squared loss $d(X_i, \check{X}_i) = (X_i - \check{X}_i)^2$. The distortion between two sequences $X^n$ and $\check{X}^n$ is then defined by $d_n(X^n, \check{X}^n) = \frac{1}{n} \sum_{i=1}^{n} (X_i - \check{X}_i)^2$, the average of the per observation distortions. We drop the subscript $n$ in $d$ when it is clear from the context. The *distortion*, or *risk*, for a $(n, B)$-rate distortion code $Q_n$ is defined as the expected loss $\mathbb{E}\, d(X^n, Q_n(X^n))$. Denoting by $\mathcal{Q}_{n,B}$ the set of all $(n, B)$-rate distortion codes, the *distortion rate function* is defined as

$$R(B, \sigma) = \liminf_{n \to \infty} \inf_{Q_n \in \mathcal{Q}_{n,B}} \mathbb{E}\, d(X^n, Q_n(X^n)).$$

This distortion rate function depends on the rate $B$ as well as the source distribution. For the i.i.d. $\mathcal{N}(0, \sigma^2)$ source, according to the well-known rate distortion theorem [7],

$$R(B, \sigma) = \sigma^2 2^{-2B}.$$

When $B$ is zero, meaning no information gets encoded at all, this bound becomes $\sigma^2$, which is the expected loss when each random variable is represented by its mean. As $B$ approaches infinity, the distortion goes to zero.

The previous discussion assumes the source random variables are independent and follow a common distribution $\mathcal{N}(0, \sigma^2)$. The goal is to minimize the expected distortion in the reconstruction of $X^n$ after transmitting or storing the data under a communication constraint. Now suppose that

$$X_i \overset{\text{ind.}}{\sim} \mathcal{N}(\theta_i, \sigma^2) \text{ for } i = 1, 2, \ldots, n.$$

We assume the variance $\sigma^2$ is known and the means $\theta^n = (\theta_1, \ldots, \theta_n)$ are unknown. Suppose, furthermore, that instead of trying to minimize the recovery distortion $d(X^n, \check{X}^n)$, we want to estimate the means with a risk as small as possible, but again using a budget of $B$ bits per index.

Without the communication constraint, this problem has been very well studied [10, 9]. Let $\widehat{\theta}(X^n) \equiv \widehat{\theta}^n = (\widehat{\theta}_1, \ldots, \widehat{\theta}_n)$ denote an estimator of the true mean $\theta^n$. For a parameter space $\Theta_n \subset \mathbb{R}^n$, the minimax risk over $\Theta_n$ is defined as

$$\inf_{\widehat{\theta}^n} \sup_{\theta^n \in \Theta_n} \mathbb{E}\, d(\theta^n, \widehat{\theta}^n) = \inf_{\widehat{\theta}^n} \sup_{\theta^n \in \Theta_n} \mathbb{E}\, \frac{1}{n} \sum_{i=1}^{n} (\theta_i - \widehat{\theta}_i)^2.$$

For the $L_2$ ball of radius $c$,

$$\Theta_n(c) = \left\{ (\theta_1, \ldots, \theta_n) : \frac{1}{n} \sum_{i=1}^{n} \theta_i^2 \leq c^2 \right\}, \tag{1}$$

Pinsker's theorem gives the exact, limiting form of the minimax risk

$$\liminf_{n \to \infty} \inf_{\widehat{\theta}^n} \sup_{\theta^n \in \Theta_n(c)} \mathbb{E}\, d(\theta^n, \widehat{\theta}^n) = \frac{\sigma^2 c^2}{\sigma^2 + c^2}.$$

To impose a communication constraint, we incorporate a variant of the source coding scheme described above into this minimax framework of estimation. Define a $(n, B)$-rate estimation code

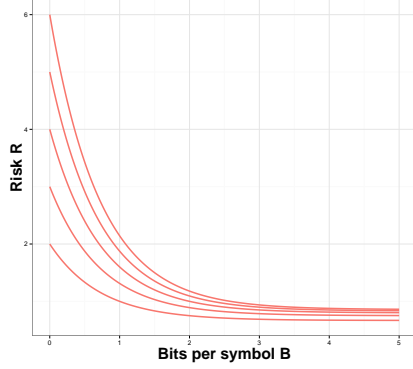

Figure 2. Our result establishes the Pareto-optimal tradeoff in the nonparametric normal means problem for risk versus number of bits:

$$R(\sigma^2, c^2, B) = \frac{c^2\sigma^2}{\sigma^2 + c^2} + \frac{c^4 2^{-2B}}{\sigma^2 + c^2}$$

Curves for five signal sizes are shown, $c^2 = 2, 3, 4, 5, 6$. The noise level is $\sigma^2 = 1$. With zero bits, the rate is $c^2$, the highest point on the risk curve. The rate for large $B$ approaches the Pinsker bound $\sigma^2 c^2/(\sigma^2 + c^2)$.

$M_n = (\phi_n, \psi_n)$, as a pair of encoding and decoding functions, as before. The encoding function $\phi_n : \mathbb{R}^n \to \{1, 2, \ldots, 2^{nB}\}$ is a mapping from observations $X^n$ to an index set. The decoding function is a mapping from indices to models $\check{\theta}^n \in \mathbb{R}^n$. We write the composition of the encoder and decoder as $M_n(X^n) = \psi_n(\phi_n(X^n)) = \check{\theta}^n$, which we call a *quantized estimator*. Denoting by $\mathcal{M}_{n,B}$ the set of all $(n, B)$-rate estimation codes, we then define the *quantized minimax risk* as

$$R_n(B, \sigma, \Theta_n) = \inf_{M_n \in \mathcal{M}_{n,B}} \sup_{\theta^n \in \Theta_n} \mathbb{E}\, d(\theta^n, M_n(X^n)).$$

We will focus on the case where our parameter space is the $L_2$ ball defined in (1), and write

$$R_n(B, \sigma, c) = R_n(B, \sigma, \Theta_n(c)).$$

In this setting, we let $n$ go to infinity and define the asymptotic quantized minimax risk as

$$R(B, \sigma, c) = \liminf_{n \to \infty} R_n(B, \sigma, c) = \liminf_{n \to \infty} \inf_{M_n \in \mathcal{M}_{n,B}} \sup_{\theta^n \in \Theta_n(c)} \mathbb{E}\, d(\theta^n, M_n(X^n)). \tag{2}$$

Note that we could estimate $\theta^n$ based on the quantized data $\check{X}^n = Q_n(X^n)$. Once again denoting by $\mathcal{Q}_{n,B}$ the set of all $(n, B)$-rate distortion codes, such an estimator is written $\check{\theta}^n = \check{\theta}^n(Q_n(X^n))$. Clearly, if the decoding functions $\psi_n$ of $\mathcal{Q}_n$ are injective, then this formulation is equivalent. The quantized minimax risk is then expressed as

$$R_n(B, \sigma, \Theta_n) = \inf_{\check{\theta}^n} \inf_{Q_n \in \mathcal{Q}_{n,B}} \sup_{\theta^n \in \Theta_n} \mathbb{E}\, d(\theta^n, \check{\theta}^n).$$

The many normal means problem exhibits much of the complexity and subtlety of general nonparametric regression and density estimation problems. It arises naturally in the estimation of a function expressed in terms of an orthogonal function basis [8, 13]. Our main result sharply characterizes the excess risk that communication constraints impose on minimax estimation for $\Theta(c)$.

## 3 Main results

Our first result gives a lower bound on the exact quantized asymptotic risk in terms of $B$, $\sigma$, and $c$.

**Theorem 1.** *For $B \geq 0$, $\sigma > 0$ and $c > 0$, the asymptotic minimax risk defined in (2) satisfies*

$$R(B, \sigma, c) \geq \frac{\sigma^2 c^2}{\sigma^2 + c^2} + \frac{c^4}{\sigma^2 + c^2} 2^{-2B}. \tag{3}$$

This lower bound on the limiting minimax risk can be viewed as the usual minimax risk without quantization, plus an excess risk term due to quantization. If we take $B$ to be zero, the risk becomes $c^2$, which is obtained by estimating all of the means simply by zero. On the other hand, letting $B \to \infty$, we recover the minimax risk in Pinsker's theorem. This tradeoff is illustrated in Figure 2.

The proof of the theorem is technical and we defer it to the supplementary material. Here we sketch the basic idea of the proof. Suppose we are able to find a prior distribution $\pi_n$ on $\theta^n$ and a random

vector $\widetilde{\theta}^n$ such that for any $(n, B)$-rate estimation code $M_n$ the following holds:

$$\frac{\sigma^2 c^2}{\sigma^2 + c^2} + \frac{c^4}{\sigma^2 + c^2} 2^{-2B} \overset{(I)}{=} \int \mathbb{E}_{X^n} d(\theta^n, \widetilde{\theta}^n) d\pi_n(\theta^n)$$

$$\overset{(II)}{\leq} \int \mathbb{E}_{X^n} d(\theta^n, M_n(X^n)) d\pi_n(\theta^n)$$

$$\overset{(III)}{\leq} \sup_{\theta^n \in \Theta_n(c)} \mathbb{E}_{X^n} d(\theta^n, M_n(X^n)).$$

Then taking an infimum over $M_n \in \mathcal{M}_{n,B}$ gives us the desired result. In fact, we can take $\pi_n$, the prior on $\theta^n$, to be $\mathcal{N}(0, c^2 I_n)$, and the model becomes $\theta_i \sim \mathcal{N}(0, c^2)$ and $X_i \mid \theta_i \sim \mathcal{N}(\theta_i, \sigma^2)$. Then according to Lemma 1, inequality (II) holds with $\widetilde{\theta}^n$ being the minimizer to the optimization problem

$$\min_{p(\widetilde{\theta}^n \mid X^n, \theta^n)} \mathbb{E}\, d(\theta^n, \widetilde{\theta}^n)$$

$$\text{subject to} \quad I(X^n; \widetilde{\theta}^n) \leq nB,$$

$$p(\widetilde{\theta}^n \mid X^n, \theta^n) = p(\widetilde{\theta}^n \mid X^n).$$

The equality (I) holds due to Lemma 2. The inequality (III) can be shown by a limiting concentration argument on the prior distribution, which is included in the supplementary material.

**Lemma 1.** *Suppose that $X_1, \ldots, X_n$ are independent and generated by $\theta_i \sim \pi(\theta_i)$ and $X_i \mid \theta_i \sim p(x_i \mid \theta_i)$. Suppose $M_n$ is an $(n, B)$-rate estimation code with risk $\mathbb{E}\, d(\theta^n, M_n(X^n)) \leq D$. Then the rate $B$ is lower bounded by the solution to the following problem:*

$$\min_{p(\widetilde{\theta}^n \mid X^n, \theta^n)} I(X^n; \widetilde{\theta}^n)$$

$$\text{subject to} \quad \mathbb{E}\, d(\theta^n, \widetilde{\theta}^n) \leq D, \tag{4}$$

$$p(\widetilde{\theta}^n \mid X^n, \theta^n) = p(\widetilde{\theta}^n \mid X^n).$$

The next lemma gives the solution to problem (4) when we have $\theta_i \sim \mathcal{N}(0, c^2)$ and $X_i \mid \theta_i \sim \mathcal{N}(\theta_i, \sigma^2)$

**Lemma 2.** *Suppose $\theta_i \sim \mathcal{N}(0, c^2)$ and $X_i \mid \theta_i \sim \mathcal{N}(\theta_i, \sigma^2)$ for $i = 1, \ldots, n$. For any random vector $\widetilde{\theta}^n$ satisfying $\mathbb{E}\, d(\theta^n, \widetilde{\theta}^n) \leq D$ and $p(\widetilde{\theta}^n \mid X^n, \theta^n) = p(\widetilde{\theta}^n \mid X^n)$ we have*

$$I(X^n; \widetilde{\theta}^n) \geq \frac{n}{2} \log \frac{c^4}{(\sigma^2 + c^2)(D - \frac{\sigma^2 c^2}{\sigma^2 + c^2})}.$$

Combining the above two lemmas, we obtain a lower bound of the risk assuming that $\theta^n$ follows the prior distribution $\pi_n$:

**Corollary 1.** *Suppose $M_n$ is a $(n, B)$-rate estimation code for the source $\theta_i \sim \mathcal{N}(0, c^2)$ and $X_i \mid \theta_i \sim \mathcal{N}(\theta_i, \sigma^2)$, then*

$$\mathbb{E}\, d(\theta^n, M_n(X^n)) \geq \frac{\sigma^2 c^2}{\sigma^2 + c^2} + \frac{c^4}{\sigma^2 + c^2} 2^{-2B}. \tag{5}$$

### 3.1 An adaptive source coding method

We now present a source coding method, which we will show attains the minimax lower bound asymptotically with high probability.

Suppose that the encoder is given a sequence of observations $(X_1, \ldots, X_n)$, and both the encoder and the decoder know the radius $c$ of the $L_2$ ball in which the mean vector lies. The steps of the source coding method are outlined below:

Step 1. Generating codebooks. The codebooks are distributed to both the encoder and the decoder.

(a) Generate codebook $\mathcal{B} = \{1/\sqrt{n}, 2/\sqrt{n}, \ldots, \lceil c^2\sqrt{n} \rceil / \sqrt{n}\}$.

(b) Generate codebook $\mathcal{X}$ which consists of $2^{nB}$ i.i.d. random vectors from the uniform distribution on the $n$-dimensional unit sphere $\mathbb{S}^{n-1}$.

Step 2. Encoding.

(a) Encode $\widehat{b}^2 = \frac{1}{n}\|X\|^2 - \sigma^2$ by $\check{b}^2 = \arg\min\{|b^2 - \widehat{b}^2| : b^2 \in \mathcal{B}\}$.

(b) Encode $X^n$ by $\check{X}^n = \arg\max\{\langle X^n, x^n \rangle : x^n \in \mathcal{X}\}$.

Step 3. Transmit or store $(\check{b}^2, \check{X}^n)$ by their corresponding indices using $\log c^2 + \frac{1}{2}\log n + nB$ bits.

Step 4. Decoding.

(a) Recover $(\check{b}^2, \check{X}^n)$ by the transmitted or stored indices.

(b) Estimate $\theta$ by

$$\check{\theta}^n = \sqrt{\frac{n\check{b}^4(1 - 2^{-2B})}{\check{b}^2 + \sigma^2}} \cdot \check{X}^n.$$

We make several remarks on this quantized estimation method.

**Remark 1.** The rate of this coding method is $B + \frac{\log c^2}{n} + \frac{\log n}{2n}$, which is asymptotically $B$ bits.

**Remark 2.** The method is probabilistic; the randomness comes from the construction of the codebook $\mathcal{X}$. Denoting by $\mathcal{M}^*_{n,B,\sigma,c}$ the ensemble of such random quantizers, there is then a natural one-to-one mapping between $\mathcal{M}^*_{n,B,\sigma,c}$ and $(\mathbb{S}^{n-1})^{2^{nB}}$ and we attach probability measure to $\mathcal{M}^*_{n,B,\sigma,c}$ corresponding to the product uniform distribution on $(\mathbb{S}^{n-1})^{2^{nB}}$.

**Remark 3.** The main idea behind this coding scheme is to encode the magnitude and the direction of the observation vector separately, in such a way that the procedure adapts to sources with different norms of the mean vectors.

**Remark 4.** The computational complexity of this source coding method is exponential in $n$. Therefore, like the Shannon random codebook, this is a demonstration of the asymptotic achievability of the lower bound (3), rather than a practical scheme to be implemented. We discuss possible computationally efficient algorithms in Section 5.

The following shows that with high probability this procedure will attain the desired lower bound asymptotically.

**Theorem 2.** *For a sequence of vectors $\{\theta^n\}_{n=1}^{\infty}$ satisfying $\theta^n \in \mathbb{R}^n$ and $\|\theta^n\|^2/n = b^2 \leq c^2$, as $n \to \infty$*

$$\mathbb{P}\left(d(\theta^n, M_n(X^n)) > \frac{\sigma^2 b^2}{\sigma^2 + b^2} + \frac{b^4}{\sigma^2 + b^2}2^{-2B} + C\sqrt{\frac{\log n}{n}}\right) \longrightarrow 0 \tag{6}$$

*for some constant $C$ that does not depend on $n$ (but could possibly depend on $b$, $\sigma$ and $B$). The probability measure is with respect to both $M_n \in \mathcal{M}^*_{n,B,\sigma,c}$ and $X^n \in \mathbb{R}^n$.*

This theorem shows that the source coding method not only achieves the desired minimax lower bound for the $L_2$ ball with high probability with respect to the random codebook and source distribution, but also adapts to the true magnitude of the mean vector $\theta^n$. It agrees with the intuition that the hardest mean vector to estimate lies on the boundary of the $L_2$ ball. Based on Theorem 2 we can obtain a uniform high probability bound for mean vectors in the $L_2$ ball.

**Corollary 2.** *For any sequence of vectors $\{\theta^n\}_{n=1}^{\infty}$ satisfying $\theta^n \in \mathbb{R}^n$ and $\|\theta^n\|^2/n \leq c^2$, as $n \to \infty$*

$$\mathbb{P}\left(d(\theta^n, M_n(X^n)) > \frac{\sigma^2 c^2}{\sigma^2 + c^2} + \frac{c^4}{\sigma^2 + c^2}2^{-2B} + C'\sqrt{\frac{\log n}{n}}\right) \longrightarrow 0$$

*for some constant $C'$ that does not depend on $n$.*

We include the details of the proof of Theorem 2 in the supplementary material, which carefully analyzes the three terms in the following decomposition of the loss function:

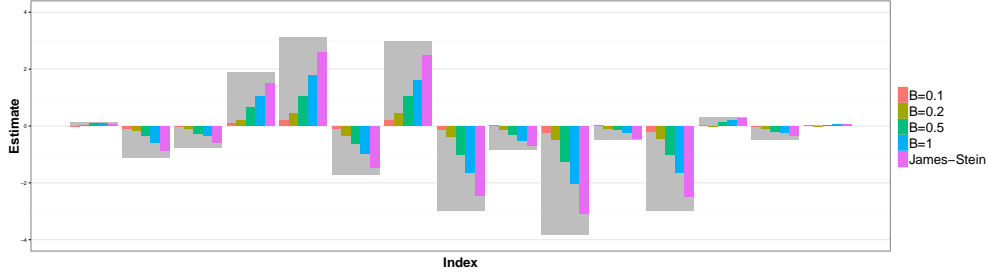

Figure 3: Comparison of the quantized estimates with different rates $B$, the James-Stein estimator, and the true mean vector. The heights of the bars are the averaged estimates based on 100 replicates. Each large background rectangle indicates the original mean component $\theta_j$.

$$d(\theta^n, \check{\theta}^n) = \frac{1}{n}\left\|\check{\theta}^n - \theta^n\right\|^2 = \frac{1}{n}\left\|\check{\theta}^n - \widehat{\gamma}X^n + \widehat{\gamma}X^n - \theta^n\right\|^2$$

$$= \underbrace{\frac{1}{n}\left\|\check{\theta}^n - \widehat{\gamma}X^n\right\|^2}_{A_1} + \underbrace{\frac{1}{n}\|\widehat{\gamma}X^n - \theta^n\|^2}_{A_2} + \underbrace{\frac{2}{n}\langle\check{\theta}^n - \widehat{\gamma}X^n, \widehat{\gamma}X^n - \theta^n\rangle}_{A_3}$$

where $\widehat{\gamma} = \frac{\widehat{b}^2}{\widehat{b}^2+\sigma^2}$ with $\widehat{b}^2 = \|X^n\|^2/n - \sigma^2$. Term $A_1$ characterizes the quantization error. Term $A_2$ does not involve the random codebook, and is the loss of a type of James-Stein estimator. The cross term $A_3$ vanishes as $n \to \infty$.

## 4    Simulations

In this section we present a set of simulation results showing the empirical performance of the proposed quantized estimation method. Throughout the simulation, we fix the noise level $\sigma^2 = 1$, while varying the other parameters $c$ and $B$.

First we show in Figure 3 the effect of quantized estimation and compare it with the James-Stein estimator. Setting $n = 15$ and $c = 2$, we randomly generate a mean vector $\theta^n \in \mathbb{R}^n$ with $\|\theta\|^2/n = c^2$. A random vector $X$ is then drawn from $\mathcal{N}(\theta^n, I_n)$ and quantized estimates with rates $B \in \{0.1, 0.2, 0.5, 1\}$ are calculated; for comparison we also compute the James-Stein estimator, given by $\widehat{\theta}_{\text{JS}}^n = \left(1 - \frac{(n-2)\sigma^2}{\|X^n\|^2}\right)X^n$. We repeat this sampling and estimation procedure 100 times and report the averaged risk estimates in Figure 3. We see that the quantized estimator essentially shrinks the random vector towards zero. With small rates, the shrinkage is strong, with all the estimates close to zero. Estimates with larger rates approach the James-Stein estimator.

In our second set of simulations, we choose $c$ from $\{0.1, 0.5, 1, 5, 10\}$ to reflect different signal-to-noise ratios, and choose $B$ from $\{0.1, 0.2, 0.5, 1\}$. For each combination of the values of $c$ and $B$, we vary $n$, the dimension of the mean vector, which is also the number of observations. Given a set of parameters $c$, $B$ and $n$, a mean vector $\theta^n$ is generated uniformly on the sphere $\|\theta^n\|^2/n = c^2$ and data $X^n$ are generated following the distribution $\mathcal{N}(\theta^n, \sigma^2 I_n)$. We quantize the data using the source coding method, and compute the mean squared error between the estimator and the true mean vector. The procedure is repeated 100 times for each of the parameter combinations, and the average and standard deviation of the mean squared errors are recorded. The results are shown in Figure 4. We see that as $n$ increases, the average error decreases and approaches the theoretic lower bound in Theorem 1. Moreover, the standard deviation of the mean squared errors also decreases, confirming the result of Theorem 2 that the convergence is with high probability.

## 5    Discussion and future work

In this paper, we establish a sharp lower bound on the asymptotic minimax risk for quantized estimators of nonparametric normal means for the case of a Euclidean ball. Similar techniques can be

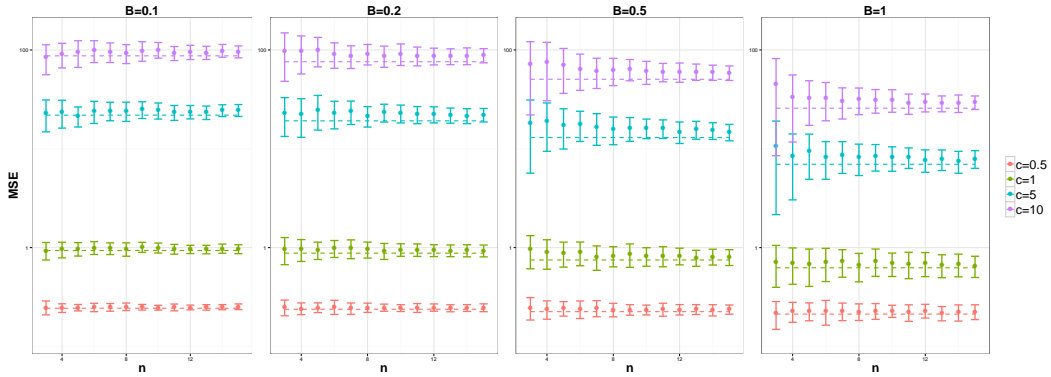

Figure 4: Mean squared errors and standard deviations of the quantized estimator versus $n$ for different values of $(B, c)$. The horizontal dashed lines indicate the lower bounds.

applied to the setting where the parameter space is an ellipsoid $\Theta = \{\theta : \sum_{j=1}^{\infty} a_j^2 \theta_j^2 \leq c^2\}$. A principal case of interest is the Sobolev ellipsoid of order $m$ where $a_j^2 \sim (\pi j)^{2m}$ as $j \to \infty$. The Sobolev ellipsoid arises naturally in nonparametric function estimation and is thus of great importance. We leave this to future work.

Donoho discusses the parallel between rate distortion theory and Pinsker's work in his Wald Lectures [4]. Focusing on the case of the Sobolev space of order $m$, which we denote by $\mathcal{F}_m$, it is shown that the Kolmogorov entropy $H_\epsilon(\mathcal{F}_m)$ and the rate distortion function $R(D, X)$ satisfy $H_\epsilon(\mathcal{F}_m) \asymp \sup\{R(\epsilon^2, X) : \mathbb{P}(X \in \mathcal{F}_m) = 1\}$ as $\epsilon \to 0$. This connects the worst-case minimax analysis and least-favorable rate distortion function for the function class. Another information-theoretic formulation of minimax rates lies in the so-called "le Cam equation" $H_\epsilon(\mathcal{F}) = n\epsilon^2$ [14, 15]. However, both are different from the direction we pursue in this paper, which is to impose communication constraints in minimax analysis.

In other related work, researchers in communications theory have studied estimation problems in sensor networks under communication constraints. Draper and Wornell [5] obtain a result on the so-called "one-step problem" for the quadratic-Gaussian case, which is essentially the same as the statement in our Corollary 1. In fact, they consider a similar setting, but treat the mean vector as random and generated independently from a known normal distribution. In contrast, we assume a fixed but unknown mean vector and establish a minimax lower bound as well as an adaptive source coding method that adapts to the fixed mean vector within the parameter space. Zhang et al. [16] also consider minimax bounds with communication constraints. However, the analysis in [16] is focused on distributed parametric estimation, where the data are distributed between several machines. Information is shared between the machines in order to construct a parameter estimate, and constraints are placed on the amount of communication that is allowed.

In addition to treating more general ellipsoids, an important direction for future work is to design computationally efficient quantized nonparametric estimators. One possible method is to divide the variables into smaller blocks and quantize them separately. A more interesting and promising approach is to adapt the recent work of Venkataramanan et al. [12] that uses sparse regression for lossy compression. We anticipate that with appropriate modifications, this scheme can be applied to quantized nonparametric estimation to yield practical algorithms, trading off a worse error exponent in the convergence rate to the optimal quantized minimax risk for reduced complexity encoders and decoders.

## Acknowledgements

Research supported in part by NSF grant IIS-1116730, AFOSR grant FA9550-09-1-0373, ONR grant N000141210762, and an Amazon AWS in Education Machine Learning Research grant. The authors thank Andrew Barron, John Duchi, and Alfred Hero for valuable comments on this work.

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
