[Supplementary Material · qne-supplement.pdf]

# A Proofs

## A.1 Proof of Theorem 1

*Proof of Lemma 1.* Denote the solution to problem (4) by $B^*(D)$. Suppose that $M_n$ is an $(n, B)$-rate estimation code $M_n$ with risk $\mathbb{E}\, d(\theta^n, M_n(X^n)) \leq D$. We have

$$B \geq I(X^n; M_n(X^n))/n \tag{7}$$
$$\geq B^*\left(\mathbb{E}\, d(\theta^n, M_n(X^n))\right) \tag{8}$$
$$\geq B^*(D), \tag{9}$$

where (7) follows from the fact that $M_n(X^n)$ takes at most $2^{nB}$ values; (8) follows from the definition of $B^*(\cdot)$; (9) follows from the monotonicity of $B^*(\cdot)$ and the fact that $\mathbb{E}\, d(\theta^n, M_n(X^n)) \leq D$. $\qquad\square$

*Proof of Lemma 2.* Suppose that $\widetilde{\theta}^n$ satisfies the conditions in problem (4). Write $\gamma = c^2/(\sigma^2 + c^2)$. For $i = 1, 2, \ldots, n$, consider the decomposition

$$
\begin{aligned}
\mathbb{E}(\theta_i - \widetilde{\theta}_i)^2 &= \mathbb{E}(\theta_i - \gamma X_i + \gamma X_i - \widetilde{\theta}_i)^2 \\
&= \mathbb{E}(\theta_i - \gamma X_i)^2 + \mathbb{E}(\widetilde{\theta}_i - \gamma X_i)^2 - 2\mathbb{E}\left((\theta_i - \gamma X_i)(\widetilde{\theta}_i - \gamma X_i)\right) \\
&= \frac{\sigma^2 c^2}{\sigma^2 + c^2} + \mathbb{E}(\widetilde{\theta}_i - \gamma X_i)^2.
\end{aligned}
$$

The last equality follows from

$$\mathbb{E}\left((\widetilde{\theta}_i - \gamma X_i)(\theta_i - \gamma X_i)\right) = \mathbb{E}\left(\mathbb{E}(\theta_i - \gamma X_i \mid X_i)\mathbb{E}(\widetilde{\theta}_i - \gamma X_i \mid X_i)\right) = 0,$$

where we have used the fact that $\theta_i \to X_i \to \widetilde{\theta}_i$ is a Markov chain and that $\mathbb{E}(\theta_i \mid X_i) = \gamma X_i$. Summing over $i = 1, \ldots, n$, we have

$$\mathbb{E}\, d(\theta^n, \widetilde{\theta}^n) = \frac{\sigma^2 c^2}{\sigma^2 + c^2} + \mathbb{E}\, d(\widetilde{\theta}^n, \gamma X^n).$$

A lower bound on the mutual information can be obtained as

$$
\begin{aligned}
\frac{1}{n}I(X^n; \widetilde{\theta}^n) = \frac{1}{n}I(\gamma X^n; \widetilde{\theta}^n) &\geq \frac{1}{n}\sum_{i=1}^{n} I(\gamma X_i; \widetilde{\theta}_i) \\
&= \frac{1}{n}\sum_{i=1}^{n}\left(h(\gamma X_i) - h(\gamma X_i \mid \widetilde{\theta}_i)\right) \\
&= \frac{1}{n}\sum_{i=1}^{n}\left(h(\gamma X_i) - h(\gamma X_i - \widetilde{\theta}_i \mid \widetilde{\theta}_i)\right) \\
&\geq \frac{1}{n}\sum_{i=1}^{n}\left(h(\gamma X_i) - h(\gamma X_i - \widetilde{\theta}_i)\right) \\
&\geq \frac{1}{n}\sum_{i=1}^{n}\left(h(\gamma X_i) - h\left(\mathcal{N}(0, \mathbb{E}(\gamma X_i - \widetilde{\theta}_i)^2)\right)\right) \tag{10} \\
&= \frac{1}{n}\sum_{i=1}^{n}\left(\frac{1}{2}\log\frac{c^4}{\sigma^2 + c^2} - \frac{1}{2}\log\mathbb{E}(\gamma X_i - \widetilde{\theta}_i)^2\right) \\
&= \frac{1}{2}\log\frac{c^4}{\sigma^2 + c^2} - \frac{1}{n}\sum_{i=1}^{n}\frac{1}{2}\log\mathbb{E}(\gamma X_i - \widetilde{\theta}_i)^2 \\
&\geq \frac{1}{2}\log\frac{c^4}{\sigma^2 + c^2} - \frac{1}{2}\log\mathbb{E}\, d(\widetilde{\theta}^n, \gamma X^n) \tag{11} \\
&= \frac{1}{2}\log\frac{\frac{c^4}{\sigma^2 + c^2}}{\mathbb{E}\, d(\theta^n, \widetilde{\theta}^n) - \frac{\sigma^2 c^2}{\sigma^2 + c^2}}
\end{aligned}
$$

where (10) follows from the fact that the normal distribution maximizes the entropy for a given second moment, (11) follows from the concavity of the $\log$ function, and the other inequalities follow from the properties of mutual information and entropy. Since $\mathbb{E}\, d(\theta^n, \widetilde{\theta}^n) \leq D$, we have

$$\frac{1}{n}I(X^n; \widetilde{\theta}^n) \geq \frac{1}{2}\log\frac{\frac{c^4}{\sigma^2+c^2}}{D - \frac{\sigma^2 c^2}{\sigma^2+c^2}}.$$

On the other hand, a calculation shows that the following joint distribution

$$\widetilde{\theta}^n \sim \mathcal{N}\left(0, \gamma^2(\sigma^2 + c^2 - D)I_n\right), \quad X^n \sim \mathcal{N}\left(\widetilde{\theta}^n/\gamma, DI_n\right), \quad \theta^n \sim \mathcal{N}(\gamma X^n, \gamma\sigma^2 I_n). \quad (12)$$

achieves the lower bound, which concludes the proof. $\qquad\square$

*Proof of Theorem 1.* Suppose that $M_n$ is an $(n, B)$-rate estimation code. Let $\pi_n$, the prior on $\theta^n$, be $\mathcal{N}(0, c^2 I_n)$. According to Lemmas 1 and 2

$$\frac{\sigma^2 c^2}{\sigma^2 + c^2} + \frac{c^4}{\sigma^2 + c^2}2^{-2B} = \int \mathbb{E}_{X^n} d(\theta^n, \widetilde{\theta}^n)d\pi_n(\theta^n)$$
$$\leq \int \mathbb{E}_{X^n} d(\theta^n, M_n(X^n))d\pi_n(\theta^n),$$

where $\widetilde{\theta}^n$ follows the distribution specified in (12). It then suffices to show that as $n \to \infty$

$$\int \mathbb{E}_{X^n} d(\theta^n, M_n(X^n))d\pi_n(\theta^n) \leq \sup_{\theta^n \in \Theta_n(c)} \mathbb{E}_{X^n} d(\theta^n, M_n(X^n)).$$

In fact, if the above inequality holds, taking a supremum over $M_n \in \mathcal{M}_{n,B}$ gives the desired lower bound. Recall that $\Theta_n(c) = \{\theta^n : \frac{1}{n}\sum_{i=1}^n \theta_i^2 \leq c^2\}$. Paralleling the argument in [9, 13], we have

$$\int \mathbb{E}_{X^n} d(\theta^n, M_n(X^n))d\pi_n(\theta^n)$$
$$= \int_{\Theta_n(c)} \mathbb{E}_{X^n} d(\theta^n, M_n(X^n))d\pi_n(\theta^n) + \int_{\overline{\Theta_n(c)}} \mathbb{E}_{X^n} d(\theta^n, M_n(X^n))d\pi_n(\theta^n)$$
$$\leq \sup_{\Theta_n(c)} \mathbb{E}_{X^n} d(\theta^n, M_n(X^n)) + \int_{\overline{\Theta_n(c)}} \mathbb{E}_{X^n} d(\theta^n, M_n(X^n))d\pi_n(\theta^n).$$

It remains to show that

$$\int_{\overline{\Theta_n(c)}} \mathbb{E}_{X^n} d(\theta^n, M_n(X^n))d\pi_n(\theta^n) \longrightarrow 0.$$

where $\overline{\Theta_n(c)}$ denotes the complement of $\Theta_n(c)$. For a fixed $\delta \in (0, 1)$, let $\pi_{n,\delta}$ be a $\mathcal{N}(0, c^2\delta^2 I_n)$ prior on $\theta^n$. Replacing $\pi_n$ by $\pi_{n,\delta}$, and using the Cauchy-Schwarz inequality, we get

$$\int_{\overline{\Theta_n(c)}} \mathbb{E}_{X^n} d(\theta^n, M_n(X^n))d\pi_{n,\delta}(\theta^n)$$
$$\leq 2\int_{\overline{\Theta_n(c)}} \frac{1}{n}\|\theta^n\|^2 d\pi_{n,\delta}(\theta^n) + 2\int_{\overline{\Theta_n(c)}} \mathbb{E}_{X^n} \frac{1}{n}\|M(X^n)\|^2 d\pi_{n,\delta}(\theta^n)$$
$$\leq \frac{2}{n}\sqrt{\pi_{n,\delta}\left(\overline{\Theta_n(c)}\right)}\sqrt{\mathbb{E}_{\pi_{n,\delta}}\|\theta^n\|^4} + 2c^2\pi_{n,\delta}\left(\overline{\Theta_n(c)}\right). \quad (13)$$

Now we bound the two terms in the formula above. First,

$$\pi_{n,\delta}\left(\overline{\Theta_n(c)}\right) = \mathbb{P}\left(\frac{1}{n}\sum_{i=1}^n \theta_i^2 > c^2\right)$$
$$= \mathbb{P}\left(\frac{1}{n}\sum_{i=1}^n \left(\left(\frac{\theta_i}{\delta c}\right)^2 - 1\right) > \frac{1 - \delta^2}{\delta^2}\right)$$
$$\leq 2\exp\left(-\frac{n(1 - \delta^2)^2}{8\delta^4}\right)$$

where the last inequality is due to the following large deviation inequality: if $Z_1, \ldots, Z_n \sim \mathcal{N}(0,1)$ and $0 < t < 1$, then

$$\mathbb{P}\left(\left|\frac{1}{n}\sum_{i=1}^{n}(Z_i^2 - 1)\right| > t\right) \leq 2e^{-nt^2/8}.$$

Next, we note that

$$\mathbb{E}_{\pi_{n,\delta}}\|\theta^n\|^4 = \sum_{i=1}^{n}\mathbb{E}_{\pi_{n,\delta}}\theta_i^4 + \sum_{i \neq j}\mathbb{E}_{\pi_{n,\delta}}\theta_i^2 \cdot \mathbb{E}_{\pi_{n,\delta}}\theta_j^2$$

$$= n\mathbb{E}_{\pi_{n,\delta}}\theta_1^4 + \binom{n}{2}c^2\delta^2$$

$$= O(n^2).$$

Therefore, we have from (13)

$$\int_{\Theta_n(c)}\mathbb{E}_{X^n}d(\theta^n, M_n(X^n))d\pi_{n,\delta}(\theta^n)$$

$$\leq \frac{2}{n}\cdot\sqrt{2}\exp\left(-\frac{n(1-\delta^2)^2}{16\delta^4}\right)O(n) + 2c^2\exp\left(-\frac{n(1-\delta^2)^2}{8\delta^4}\right) \longrightarrow 0$$

for any $\delta \in (0,1)$. The conclusion then follows by letting $\delta \uparrow 1$. $\qquad\square$

## A.2 Proof of Theorem 2

*Proof of Theorem 2.* Suppose that $\|\theta^n\|/n = b^2 \leq c^2$ and that $X_i \sim \mathcal{N}(\theta_i, \sigma^2)$. Writing $\widehat{b}^2 = \|X^n\|^2/n - \sigma^2$ and $\widehat{\gamma} = \frac{\widehat{b}^2}{\sigma^2 + \widehat{b}^2}$, we have the decomposition of the loss

$$d(\theta^n, \check{\theta}^n) = \frac{1}{n}\left\|\check{\theta}^n - \theta^n\right\|^2$$

$$= \frac{1}{n}\left\|\check{\theta}^n - \widehat{\gamma}X^n + \widehat{\gamma}X^n - \theta^n\right\|^2$$

$$= \underbrace{\frac{1}{n}\left\|\check{\theta}^n - \widehat{\gamma}X^n\right\|^2}_{A_1} + \underbrace{\frac{1}{n}\left\|\widehat{\gamma}X^n - \theta^n\right\|^2}_{A_2} + \underbrace{\frac{2}{n}\langle\check{\theta}^n - \widehat{\gamma}X^n, \widehat{\gamma}X^n - \theta^n\rangle}_{A_3}.$$

(i) Term $A_1$ characterizes the quantization error. It has the following decomposition

$$A_1 = \frac{1}{n}\|\check{\theta}^n\|^2 + \frac{1}{n}\widehat{\gamma}^2\|X^n\|^2 - \frac{2}{n}\langle\check{\theta}^n, \widehat{\gamma}X^n\rangle.$$

By Lemma 3 and Lemma 5 below, we have

$$\frac{1}{n}\|X^n\|^2 - b^2 - \sigma^2 = O_P\left(\frac{1}{\sqrt{n}}\right), \qquad \frac{\langle X^n, \check{X}^n\rangle}{\|X\|} - \sqrt{1 - 2^{-2B}} = O_P\left(\frac{\log n}{n}\right),$$

and therefore

$$\frac{1}{n}\|\check{\theta}^n\|^2 = \frac{\check{b}^4(1 - 2^{-2B})}{\sigma^2 + \check{b}^2} = \frac{b^4}{\sigma^2 + b^2}(1 - 2^{-2B}) + O_P\left(\frac{1}{\sqrt{n}}\right),$$

$$\frac{1}{n}\widehat{\gamma}^2\|X^n\|^2 = \frac{\widehat{b}^4}{\sigma^2 + \widehat{b}^2} = \frac{b^4}{\sigma^2 + b^2} + O_P\left(\frac{1}{\sqrt{n}}\right),$$

$$\frac{2}{n}\langle\check{\theta}^n, \widehat{\gamma}X^n\rangle = \frac{2}{n}\frac{\widehat{b}^2}{\sigma^2 + \widehat{b}^2}\sqrt{\frac{n\check{b}^4(1 - 2^{-2B})}{\sigma^2 + \check{b}^2}}\cdot\langle X^n, \check{X}^n\rangle$$

$$= \frac{2\widehat{b}^2\check{b}^2\sqrt{1 - 2^{-2B}}}{\sqrt{(\sigma^2 + \widehat{b}^2)(\sigma^2 + \check{b}^2)}}\frac{\langle X, \check{X}\rangle}{\|X\|}$$

$$= \frac{2b^4}{\sigma^2 + b^2}(1 - 2^{-2B}) + O_P\left(\frac{1}{\sqrt{n}}\right).$$

which, combined together, gives us

$$A_1 = \frac{b^4}{\sigma^2 + b^2}(1 - 2^{-2B}) + \frac{b^4}{\sigma^2 + b^2} - \frac{2b^4}{\sigma^2 + b^2}(1 - 2^{-2B}) + O_P\left(\frac{1}{\sqrt{n}}\right)$$

$$= \frac{b^4}{\sigma^2 + b^2}2^{-2B} + O_P\left(\frac{1}{\sqrt{n}}\right).$$

(ii) Term $A_2$ does not involve random codebook, and is essentially the average loss of a James-Stein-type estimator. Suppose that $A_n$ is an $n \times n$ orthonormal matrix such that $A_n\theta^n = (\sqrt{n}b, 0, \ldots, 0)^T$, which we will denote by $\tau^n$. Let $Y^n = A_nX^n$. Then $Y^n \sim \mathcal{N}(\tau^n, \sigma^2 I)$ and $\|X^n\| = \|Y^n\|$. Expressing $A_2$ in terms of $Y^n$, we have

$$A_2 = \frac{1}{n}\|\widehat{\gamma}X^n - \theta^n\|^2$$

$$= \frac{1}{n}\|\widehat{\gamma}A_nX^n - A_n\theta^n\|^2$$

$$= \frac{1}{n}\|\widehat{\gamma}Y^n - \tau^n\|^2$$

$$= \frac{1}{n}\widehat{\gamma}^2\|Y^n\|^2 - 2\widehat{\gamma}b\frac{Y_1}{\sqrt{n}} + b^2$$

$$= \frac{b^2\sigma^2}{\sigma^2 + b^2} + O_P\left(\frac{1}{\sqrt{n}}\right).$$

The last equality is because

$$\frac{1}{n}\|Y^n\|^2 - b^2 - \sigma^2 = O_P\left(\frac{1}{\sqrt{n}}\right), \quad \frac{Y_1}{\sqrt{n}} - b = O_P\left(\frac{1}{\sqrt{n}}\right).$$

(iii) Finally, it can be shown that the cross term satisfies $A_3 = O_P(\sqrt{\frac{\log n}{n}})$ by exploiting the geometry of the vectors, and using the fact that most vectors are nearly orthogonal to each other in a high dimensional space. In fact, write

$$\widehat{\theta}^n = \sqrt{\frac{n\widehat{b}^4(1 - 2^{-2B})}{\widehat{b}^2 + \sigma^2}} \cdot \check{X}^n.$$

Then in the decomposition

$$\frac{2}{n}\langle\check{\theta}^n - \widehat{\gamma}X^n, \widehat{\gamma}X^n - \theta^n\rangle = \frac{2}{n}\langle\check{\theta}^n - \widehat{\theta}^n, \widehat{\gamma}X^n - \theta^n\rangle + \frac{2}{n}\langle\widehat{\theta}^n, \widehat{\gamma}X^n - \theta^n\rangle - \frac{2}{n}\langle\widehat{\gamma}X^n, \widehat{\gamma}X^n - \theta^n\rangle,$$

the first term is $O_P(\frac{1}{\sqrt{n}})$, since $\frac{1}{\sqrt{n}}\|\check{\theta}^n - \widehat{\theta}^n\| = O_P(\frac{1}{\sqrt{n}})$ and $\frac{1}{\sqrt{n}}\|\widehat{\gamma}X^n - \theta^n\|$ has bounded second moment, and the third term is

$$\frac{2}{n}\langle\widehat{\gamma}X^n, \widehat{\gamma}X^n - \theta^n\rangle = \frac{2}{n}\langle\widehat{\gamma}Y^n, \widehat{\gamma}Y^n - \tau^n\rangle$$

$$= \frac{2}{n}\widehat{\gamma}^2\|Y^n\|^2 - 2\widehat{\gamma}b\frac{Y_1}{\sqrt{n}} = O_P\left(\frac{1}{\sqrt{n}}\right).$$

Now consider the second term

$$\frac{2}{n}\langle\widehat{\theta}^n, \widehat{\gamma}X^n - \theta^n\rangle = \frac{2}{n}\langle\widehat{\theta}^n, \widetilde{\gamma}X^n - \theta^n\rangle + \frac{2}{n}(\widehat{\gamma} - \widetilde{\gamma})\langle\widehat{\theta}^n, X^n\rangle$$

where

$$\widetilde{\gamma} \triangleq \frac{\sum_{i=1}^n \theta_i X_i}{\|X^n\|^2}, \quad \text{satisfying } \frac{2}{n}(\widehat{\gamma} - \widetilde{\gamma})\langle\widehat{\theta}^n, X^n\rangle = O_P\left(\frac{1}{\sqrt{n}}\right) \text{ and } \langle X^n, \widetilde{\gamma}X^n - \theta^n\rangle = 0.$$

Thus, we are left with one last term to analyze:

$$\frac{2}{n}\langle \widehat{\theta}^n, \widetilde{\gamma}X^n - \theta^n\rangle = \frac{2}{n}\sqrt{\frac{n\widehat{b}^4(1-2^{-2B})}{\widehat{b}^2 + \sigma^2}}\langle \check{X}^n, \widetilde{\gamma}X^n - \theta^n\rangle$$

$$= \sqrt{\frac{4\widehat{b}^4(1-2^{-2B})}{\widehat{b}^2 + \sigma^2}}\langle \check{X}^n, \frac{1}{\sqrt{n}}(\widetilde{\gamma}X^n - \theta^n)\rangle$$

The scaling factor in front of the inner product is some constant plus an $O_P(\frac{1}{\sqrt{n}})$ term, so we consider the inner product. Notice that the projection of $\check{X}^n$ onto the orthogonal space of $X^n$, $\mathrm{Proj}_{X^{n\perp}}(\check{X}^n)$, is independent of $X^n$. Furthermore, by symmetry, $\mathrm{Proj}_{X^{n\perp}}(\check{X}^n)$ has a spherical distribution in $\mathbb{R}^{n-1}$, and has a length $\sqrt{1-2^{-2B}} + O_P(\frac{\log n}{n})$. That is, we can write

$$\mathrm{Proj}_{X^{n\perp}}(\check{X}^n) = L_n \cdot U^n$$

where $U^n$ follows the uniform distribution on sphere $\mathbb{S}^{n-1}$ and $L_n = \sqrt{1-2^{-2B}} + O_P(\frac{\log n}{n})$. Conditioning on $X^n$, since $\langle X^n, \widetilde{\gamma}X^n - \theta^n\rangle = 0$, we have

$$\mathbb{P}\left(\langle \check{X}^n, \frac{1}{\sqrt{n}}(\widetilde{\gamma}X^n - \theta^n)\rangle > t \mid X^n = x^n\right)$$

$$= \mathbb{P}\left(\langle \mathrm{Proj}_{X^{n\perp}}(\check{X}^n), \mathrm{Proj}_{X^{n\perp}}(\frac{1}{\sqrt{n}}(\widetilde{\gamma}X^n - \theta^n))\rangle > t \mid X^n = x^n\right)$$

$$= \mathbb{P}\left(L_n\|\frac{1}{\sqrt{n}}(\widetilde{\gamma}x^n - \theta^n)\|\langle U^n, e^n\rangle > t\right)$$

$$\leq K_1\sqrt{n}\left(1 - \frac{t^2}{K_2\|\frac{1}{\sqrt{n}}(\widetilde{\gamma}x^n - \theta^n)\|^2}\right)^{\frac{n-2}{2}}.$$

where $K_1$ and $K_2$ are positive constants, and the last inequality follows from Lemma 6 below. It then follows that

$$\mathbb{P}\left(\langle \check{X}^n, \frac{1}{\sqrt{n}}(\widetilde{\gamma}X^n - \theta^n)\rangle > t\right)$$

$$= \int \mathbb{P}\left(\langle \check{X}^n, \frac{1}{\sqrt{n}}(\widetilde{\gamma}X^n - \theta^n)\rangle > t \mid X^n = x^n\right) p_{X^n}(x^n)dx^n$$

$$\leq K_1\sqrt{n}\int \left(1 - \frac{t^2}{K_2\|\frac{1}{\sqrt{n}}(\widetilde{\gamma}x^n - \theta^n)\|^2}\right)^{\frac{n-2}{2}} p_{X^n}(x^n)dx^n$$

$$\leq K_1\sqrt{n}\left(1 - \frac{t^2}{K_2'}\right)^{\frac{n-2}{2}} + \mathbb{P}\left(\frac{1}{n}\|X^n\|^2 > b^2 + \sigma^2 + K_3\right).$$

for positive constants $K_1$, $K_2$ and $K_3$. This implies that $\langle \check{X}^n, \frac{1}{\sqrt{n}}(\widetilde{\gamma}X^n - \theta^n)\rangle = O_P(\sqrt{\frac{\log n}{n}})$ and thus $A_3 = O_P(\sqrt{\frac{\log n}{n}})$.

Combining the above analyses for $A_1$, $A_2$ and $A_3$ together gives us the theorem. $\qquad\square$

**Lemma 3.** *Suppose that $X_i \overset{ind.}{\sim} N(\theta_i, \sigma^2)$, for $i = 1, \ldots, n$ and that $\frac{1}{n}\sum_{i=1}^{n}\theta_i^2 = b^2$. Then*

$$\mathbb{P}\left(\left|\frac{1}{n}\sum_{i=1}^{n}X_i^2 - b^2 - \sigma^2\right| \geq t\right) \leq 2\exp\left(-\frac{nt^2}{32\sigma^4}\right) + \frac{8\sigma b}{\sqrt{2\pi nt^2}}\exp\left(-\frac{nt^2}{32\sigma^2 b^2}\right).$$

*Specifically, if we write $\widehat{b}^2 = \|X\|^2/n - \sigma^2$, we have $\widehat{b}^2 - b^2 = O_P(\frac{1}{\sqrt{n}})$.*

*Proof.* Writing $X_i = \theta_i + \epsilon_i$, we have

$$\mathbb{P}\left(\left|\frac{1}{n}\sum_{i=1}^n X_i^2 - b^2 - \sigma^2\right| > t\right)$$

$$= \mathbb{P}\left(\left|\frac{1}{n}\sum_{i=1}^n (\epsilon_i^2 - \sigma^2) + \frac{2}{n}\sum_{i=1}^n \theta_i\epsilon_i\right| > t\right)$$

$$\leq \mathbb{P}\left(\left|\frac{1}{n}\sum_{i=1}^n (\epsilon_i^2 - \sigma^2)\right| > \frac{t}{2}\right) + \mathbb{P}\left(\left|\frac{2}{n}\sum_{i=1}^n \theta_i\epsilon_i\right| > \frac{t}{2}\right)$$

$$\leq \mathbb{P}\left(\left|\frac{1}{n}\sum_{i=1}^n \left(\left(\frac{\epsilon_i}{\sigma}\right)^2 - 1\right)\right| > \frac{t}{2\sigma^2}\right) + \mathbb{P}\left(\left|\frac{2}{n}\mathcal{N}(0, n\sigma^2 b^2)\right| > \frac{t}{2}\right)$$

$$\leq 2\exp\left(-\frac{nt^2}{32\sigma^4}\right) + \frac{8\sigma b}{\sqrt{2\pi n t^2}}\exp\left(-\frac{nt^2}{32\sigma^2 b^2}\right)$$

where the last inequality follows from the previously mentioned large deviation inequality and the upper tail inequality for the normal distribution. $\square$

**Lemma 4** (Lemma 4.1 from [2]). *Suppose that $Y$ is uniformly distributed on the $n$-dimensional unit sphere $\mathbb{S}^{n-1}$. For $x \in \mathbb{R}^n$ such that $\|x\|_2 = 1$, the inner product $\rho = \langle x, Y\rangle$ between $x$ and $Y$ has density function*

$$f(\rho) = \frac{1}{\sqrt{\pi}}\frac{\Gamma(\frac{n}{2})}{\Gamma(\frac{n-1}{2})}(1-\rho^2)^{\frac{n-3}{2}}I(|\rho| < 1).$$

**Lemma 5.** *Suppose that $p = e^{n\beta}$ and $Y_1, \ldots, Y_p$ are independent and identically distributed with a uniform distribution on the $n$-dimensional sphere $\mathbb{S}^{n-1}$. For a fixed unit vector $x \in \mathbb{R}^n$, let $\rho_i = \langle x, Y_i\rangle$ and $L_n = \max_{1\leq i \leq p} \rho_i$. Then $L_n \to \sqrt{1-e^{-2\beta}}$ in probability as $n \to \infty$. Furthermore, $L_n - \sqrt{1-e^{-2\beta}} = O_P(\frac{\log n}{n})$.*

*Proof.* Let $k_n = \frac{n}{\log n}$. For any fixed $u \in \mathbb{R}$

$$\mathbb{P}\left(k_n\left(L_n - \sqrt{1-e^{-2\beta}}\right) \leq u\right)$$

$$= \mathbb{P}\left(L_n \leq \frac{u}{k_n} + \sqrt{1-e^{-2\beta}}\right)$$

$$= \mathbb{P}\left(\rho_1 \leq \frac{u}{k_n} + \sqrt{1-e^{-2\beta}}\right)^p$$

$$= \left(1 - \int_{u/k_n+\sqrt{1-e^{-2\beta}}}^1 \frac{1}{\sqrt{\pi}}\frac{\Gamma(\frac{n}{2})}{\Gamma(\frac{n-1}{2})}(1-\rho^2)^{\frac{n-3}{2}}d\rho\right)^p$$

$$\sim \left(1 - \frac{\sqrt{n}}{\sqrt{2\pi}(n-3)(\frac{u}{k_n}+\sqrt{1-e^{-2\beta}})}\left(1 - \left(\frac{u}{k_n}+\sqrt{1-e^{-2\beta}}\right)^2\right)^{\frac{n-1}{2}}\right)^p$$

$$\sim \exp\left(-p \cdot \frac{\sqrt{n}}{\sqrt{2\pi}(n-3)(\frac{u}{k_n}+\sqrt{1-e^{-2\beta}})}\left(1 - \left(\frac{u}{k_n}+\sqrt{1-e^{-2\beta}}\right)^2\right)^{\frac{n-1}{2}}\right)$$

$$\triangleq \exp(-M).$$

Taking the logarithm of the exponent $M$, we get

$$\log M = \log p + \frac{n-1}{2}\log\left(1 - \left(\frac{u}{k_n}+\sqrt{1-e^{-2\beta}}\right)^2\right) + \log\frac{\sqrt{n}}{\sqrt{2\pi}(n-3)(\frac{u}{k_n}+\sqrt{1-e^{-2\beta}})}$$

$$\sim n\beta + \frac{n-1}{2}\log\left(e^{-2\beta} - \frac{u^2}{k_n^2} - \frac{2u}{k_n}\sqrt{1-e^{-2\beta}}\right) - \frac{1}{2}\log n - \frac{1}{2}\log\left(1-e^{-2\beta}\right) - \frac{1}{2}\log(2\pi)$$

$$\sim n\beta - (n-1)\beta - \frac{n-1}{2}\frac{u^2}{k_n^2} - (n-1)\frac{u}{k_n}\sqrt{1-e^{-2\beta}} - \frac{1}{2}\log n - \frac{1}{2}\log\left(1-e^{-2\beta}\right) - \frac{1}{2}\log(2\pi)$$

$$\sim \beta - \left(u\sqrt{1-e^{-2\beta}} + \frac{1}{2}\right)\log n - \frac{1}{2}\log\left(1-e^{-2\beta}\right) - \frac{1}{2}\log(2\pi).$$

If $u > 0$, then as $n \to \infty$, $M \to 0$, and thus $\mathbb{P}\left(k_n\left(L_n - \sqrt{1-e^{-2\beta}}\right) \leq u\right) \to 0$. If $u < -\frac{1}{2\sqrt{1-2^{-2\beta}}}$, then as $n \to \infty$, $M \to \infty$, and hence $\mathbb{P}\left(k_n\left(L_n - \sqrt{1-e^{-2\beta}}\right) \leq u\right) \to 1$. We can then conclude that $|L_n - \sqrt{1-e^{-2\beta}}| = O_P(\frac{\log n}{n})$. $\qquad\square$

**Lemma 6.** *Let $U$ have a uniform distribution on the unit sphere $\mathbb{S}^{n-1}$ and let $x \in \mathbb{R}^n$ be a fixed vector. Then*

$$\mathbb{P}\left(|\langle U, x\rangle| > \epsilon\right) \leq K\sqrt{n}(1-\epsilon^2)^{\frac{n-2}{2}}.$$

*for all $n \geq 2$ and $\epsilon \in (0,1)$, where $K$ is a universal constant. Therefore,*

$$\langle U, x\rangle = O_P\left(\sqrt{\frac{\log n}{n}}\right).$$

*Proof.* This is a direct result from Proposition 1 in [1]. $\qquad\square$