[Reviews · NeurIPS 2014]

Submitted by Assigned_Reviewer_21

The paper derives a version of Pinsker's theorem for the normal means model (minimax lower bound), when a quantized version of the data is observed rather than the original data. It also provides a strategy for encoding and decoding (estimating the mean parameters from the quantized data) that achieves the lower bound.

The paper is written clearly and the result are certainly useful from a practical point of view too. I went over the proofs and they seem correct to me. As the authors acknowledge, the encoding-decoding method is computationally prohibitive. It would be interesting to see if the results of [12] (and a sequence of follow up/related work) could be leveraged in this context to design a computationally efficient schemes.

Lastly, I am not particularly in favor of the title quantized 'non-parametric estimation'. I feel quantized 'infinite Gaussian sequence model' or 'normal means model' might be more apt for this work. The reason being, there is no rigorous proof of the equivalence (in a LeCam sense, along the lines of Brown and Low, 1996 for the unquantized case) between the quantized normal means model and non-parametric estimation problems in this paper.
Summary: The paper derives a version of Pinsker's theorem for the normal means model (minimax lower bound), when a quantized version of the data is observed rather than the original data. The paper is written clearly and provides significant theoretical insights for a practically relevant problem.

Submitted by Assigned_Reviewer_28

The authors consider Pinsker's theorem on the minimax rate in nonparametric estimation with squared error risk of normal means in an L2 ball. They extend this theorem to a rate-distortion-like setting, where there is a bound on the number of bits used in estimation. For bitrates (nr of bits used per outcome) bounded by B, the authors derive exact formulas for the n -> infinity limit of the minimax risk. As one would expect, the resulting formula becomes Pinsker's formula as B -> infinity, and degenerates to the radius of the L2 ball as B -> 0.

QUALITY

The paper seems technically sound (though I did not check all details).
While the techniques used are, in the end, variations on standard techniques in rate distortion theory (Theorem 2) and 'establishing-minimax risk lower bounds'-literature (Theorem 1), the specific details are quite different from earlier works and seem pretty nontrivial, in fact quite difficult.

CLARITY

The paper is really quite well-written. Inevitably it requires some prior knowledge on the readers' part but the authors did their best I think.

ORIGINALITY

To the best of my knowledge (note though that I am not very familiar with the recent literature) the results presented here are new.
Still (and this is my one qualm with the paper), the techniques on which they build are not new at all, and little insight is given in how much of the current work is already 'almost known' or 'rather to be expected' given earlier work. In this respect I think that the third paragraph in Section 5, Discussion, is really far too restricted and, should the paper be accepted, should be much elaborated on - even if this means cutting some other material.
In fact, I checked the Yang and Barron (YB) paper quoted by the authors in Section 5 and found that the connections seem to run deeper than suggested. Specifically, the YB proof of the YB Theorem 1 is quite similar to the line of reasoning followed by the authors to prove the author's Theorem 1. Let me elaborate on this: while in the classical minimax risk lower bounds as established by YB, there are indeed no constraints on the nr of bits used to store/communicate the estimator (so in this sense the authors are correct that they 'pursue a different direction'), YB's *proof* of such unconstrained bounds does proceed by first looking at restricted set of potential 'truths' (one puts an \epsilon-net on the set \Theta_n) and putting a discrete prior w on them, and giving (via Fano's inequality) a lower bound on the risk of the estimators which are optimal relative to w. Such estimators will *automatically* be representable in a restricted nr of bits (log of the size of the \epsilon-net, to be precise), and thus automatically satisfy a constraint of the type the authors impose. This leads one to wonder whether there isn't already 'implicit rate distortion' going on in classical minimax risk lower bounds, and whether the results in the present paper cannot be more easily re-derived as simple corollaries of existing results. I found it strange that none of this is discussed in the paper.

So all in all I would say that the *details* of the paper are certainly original - the result is new, the precise proofs are new and highly nontrivial, but the 'general type of result' is less new than the authors suggest; this is not a reason for rejection per se but it definitely needs more attention.

SIGNIFICANCE

For the same reasons I gave as to the originality, I think that while there is reasonable significance, it is not perhaps as high as the paper suggests.

PRO/CON-Analysis

PRO: well-written, highly nontrivial and useful result.
CONTRA: builds heavily on earlier work, and precise relation to it is not discussed very well

COMMENT ON IMPACT SCORE: I had a hard time choosing, I'm not really happy there are just two options here. In the end I chose 'incremental' because, while the setting is different and quite novel (marrying rate distortion and minimax nonparametric analysis), the way this is analyzed is *very* similar to existing methods in both fields ; and that these methods are interrelated was already quite well-known. So, mathematically speaking, the term 'incremental' seems justified.
Summary: The paper combines ideas from rate-distortion theory with minimax bounds for nonparametric inference. The result seems technically sound and is nontrivial, its strength being that it gives precise asymptotics, its weakness being that the fact that 'something like this' would be the case is not very surprising.

Submitted by Assigned_Reviewer_30

The authors derive a lower-bound on minimax risk for the estimation of normal means under a "communication constraint": a restriction on the number of bits that can be used to represent a given estimate. The authors devise a theoretical (not practically implementable) discretized estimator that achieves this bound, and show performance results in small-scale simulations.

The paper is quite well written, and the presentation is clear and organized. The results are interesting and, to the best of my knowledge, novel.

The authors do not propose a practical encoding method that achieves their bound, nor do they evaluate known encoding methods with respect to their bound. Further, the model used to derive the bound could be generalized significantly; the results do not seem to have immediate practical application, which limits its scope somewhat. Nevertheless, the paper is an interesting step toward understanding the tradeoff between statistical power and storage resources.
Summary: In this interesting, well written paper, the authors derive a lower-bound on minimax risk for the estimation of normal means under a "communication constraint": a restriction on the number of bits that can be used to represent a given estimate. The work is a worthwhile theoretical advance toward understanding the tradeoff between statistical power and storage resources.
Author Feedback
Author rebuttal: We thank the reviewers for their helpful comments. We would like to respond to the points raised:

We agree with Reviewer 21's comment about computationally efficient encoding/decoding schemes. Methods in [12] based on sparse blockwise regression are promising; this is an exciting future research direction.

Reviewer 21 comments on our choice of title, and the formal equivalence between the normal means model and nonparametric estimation problems. Reviewer 30 also comments on the generality of the results. We'd like to point out that our proof technique for the Euclidean ball case lays the foundation for more general estimation problems, including Sobolev ellipsoids. Indeed, the Euclidean ball case is used in known approaches to adaptive estimation for ellipsoids, where the sequence is broken up into blocks, with each block treated as a separate Euclidean ball. We have obtained results for ellipsoids, but restricting to the Euclidean ball case helps keep the current paper conceptually and technically clean and self-contained. But perhaps our title is too generic, and could be changed.

Reviewer 28 remarks that the discussion section is too restricted, and makes some interesting comments on the results in the cited Yang and Barron paper [15]. As we point out, a strong connection and parallel between coding theory and Kolmogorov epsilon-entropy has been known. This is discussed in [15] and in more depth in Donoho's Wald lectures [4]. However, the subtlety and technical challenge in the problem we formulate is to simultaneously control the quantization error and the estimation error. This requires a combination of rate-distortion and minimax estimation methods that---to the best of our knowledge---has not been addressed before.

Reviewer 28 remarks "the fact that 'something like this' would be the case is not very surprising," and views this as a weakness of the paper. Considering the incisive minds that have studied nonparametric estimation and rate distortion separately, we view it as a strength of the paper that such theorems were not previously formulated! In any case, we agree that these connections deserve a more extended discussion, and we will try to squeeze more in.